# Identification of Genetic Alterations in Rapid Progressive Glioblastoma by Use of Whole Exome Sequencing

**DOI:** 10.3390/diagnostics13061017

**Published:** 2023-03-07

**Authors:** Imran Khan, Esra Büşra Işık, Sadaf Mahfooz, Asif M. Khan, Mustafa Aziz Hatiboglu

**Affiliations:** 1Department of Molecular Biology, Beykoz Institute of Life Sciences and Biotechnology, Bezmialem Vakif University, Beykoz, 34820 Istanbul, Turkey; 2Department of Microbiology, Beykoz Institute of Life Sciences and Biotechnology, Bezmialem Vakif University, Beykoz, 34820 Istanbul, Turkey; 3Centre for Bioinformatics, School of Data Sciences, Perdana University, Damansara Heights, Kuala Lumpur 50490, Malaysia; 4Department of Neurosurgery, Bezmialem Vakif University Medical School, Vatan Street, Fatih, 34093 Istanbul, Turkey

**Keywords:** whole exome sequencing, glioblastoma, cancer, tumor mutation burden, genetic screening, genetic signature

## Abstract

Background: Glioblastoma poses an inevitable threat to patients despite aggressive therapy regimes. It displays a great level of molecular heterogeneity and numerous substitutions in several genes have been documented. Next-generation sequencing techniques have identified various molecular signatures that have led to a better understanding of the molecular pathogenesis of glioblastoma. In this limited study, we sought to identify genetic variants in a small number of rare patients with aggressive glioblastoma. Methods: Five tumor tissue samples were isolated from four patients with rapidly growing glioblastoma. Genomic DNA was isolated and whole exome sequencing was used to study protein-coding regions. Generated FASTQ files were analyzed and variants were called for each sample. Variants were prioritized with different approaches and functional annotation was applied for the detrimental variants. Results: A total of 49,780 somatic variants were identified in the five glioblastoma samples studied, with the majority as missense substitutions. The top ten genes with the highest number of substitutions were *MUC3A*, *MUC4*, *MUC6*, *OR4C5*, *PDE4DIP*, *AHNAK2*, *OR4C3*, *ZNF806*, *TTN*, and *RP1L1*. Notably, variant prioritization after annotation indicated that the *MTCH2* (Chr11: 47647265 A>G) gene sequence change was putative deleterious in all of the aggressive tumor samples. Conclusion: The *MTCH2* (Chr11: 47647265 A>G) gene substitution was identified as putative deleterious in highly aggressive glioblastomas, which merits further investigation. Moreover, a high tumor mutation burden was observed, with a signature of the highest substitutions in *MUC3A*, *MUC4*, *MUC6*, *OR4C5*, *PDE4DIP*, *AHNAK2*, *OR4C3*, *ZNF806*, *TTN*, and *RP1L1* genes. The findings provide critical, initial data for the further rational design of genetic screening and diagnostic approaches against aggressive glioblastoma.

## 1. Introduction

Glioblastoma, a rare cancer, is the most common type of malignant brain cancer, arising from glial cells, and accounts for approximately 16,606 annual deaths in the United States [1]. Despite the multimodal treatment strategy, including surgical resection followed by standard concomitant radiotherapy and chemotherapy, patients have a poor prognosis with short-term overall survival (OS) and progression-free survival [2,3]. Only a small subset of patients benefits from the currently available therapies. The main reasons proposed for this selective response are either the inability of the chemotherapeutic agents to penetrate through the blood–brain barrier (BBB) or the molecular heterogeneity of glioblastomas [4]. Molecular heterogeneity plays a crucial role in glioblastomas, not only between patients but also within the same tumor (intratumoral) [5,6,7]. Earlier classification of gliomas was primarily based on the histopathological features of the tumor [8]. Glioblastoma tumors are also known as “multiforme” owing to their gross appearance, which includes a necrotic inner core surrounded by highly anaplastic cells displaying differential levels of vascularization [9].

Several genomic studies have reported various somatic variations and epigenetic modifications in human cancers [10,11]. The applications of next-generation sequencing (NGS) techniques have helped identify several molecular targets, such as genes and specific substitutions that are crucial for glioblastoma development and progression [12,13,14]. Consequently, several potential diagnostics, prognostic, and predictive markers have been identified. These markers may help precise tumor classification, identify the risk of tumor recurrence, and better understand the complexity of the tumor [4]. Cancer Genome Atlas Research Network (TCGA) documents a plethora of genes associated with glioblastoma, including those observed with high mutations, such as isocitrate dehydrogenase (*IDH*), phosphatase and tensin homolog (*PTEN*), O6-methylguanine DNA methyltransferase (*MGMT*), tumor suppressor protein (*TP53)*, epidermal growth factor receptor (*EGFR*), vascular endothelial growth factor (*VEGF*), and *p16INK4a*. Most of these genes are utilized as a part of routine clinical examinations of patients with glioblastoma [4]. Considering the importance of genetic markers in the diagnosis and prognosis of glioblastoma patients, the recent classification of the central nervous system (CNS) tumors has shifted towards a molecular-marker-based classification system [15].

A primary glioblastoma tumor can evolve into a secondary recurring one in the presence of therapy, which can result in selection pressure giving rise to different sets of cancer cells with different molecular markers [16]. The evolution of glioblastoma tumor recurrence can be explained using either a linear or branched model. The former is characterized by high retention of primary tumor mutations [17], while the latter leads to a low carryover of mutations from primary tumors [18]. Low-grade glioma tumors and their corresponding recurrent tumors display varying genetic signatures, even for driver mutations [19]. A subset of recurring glioblastomas can show aggressive behavior and a poorer outcome.

Targeted therapies on recurrent glioblastomas based on genetic signatures detected upon diagnosis have a high tendency to fail [20]. While glioblastomas have been explored extensively at the molecular level, there is still limited data on the molecular evaluation of aggressive glioblastoma patients who exhibit rapid regrowth and resistance to conventional therapies. Herein, the genetic makeup of glioblastoma patients with rapid progression was investigated via whole exome sequencing. Though the study involves a limited number of patients (given the rarity of glioblastoma), it provides additional critical data points, much needed to better understand aggressive glioblastomas.

## 2. Materials and Methods

### 2.1. Patient Sample

Five tumor tissues were used for this study, isolated from four patients who underwent surgical resection of glioblastoma and/or received related therapy at the department of Neurosurgery, Bezmialem Vakif University. The purpose was to identify the markers against glioblastomas for rapid growth and resistance to conventional therapies. These patients were selected using the criteria of tumor growth within three months after an aggressive treatment or initial diagnosis. All patients underwent gross-total resection of the tumor and/or received radiation and chemotherapy with Temozolomide. Histological evaluation of the tumors revealed a glioblastoma World Health Organization (WHO) classification of grade IV. Despite aggressive treatment, these patients did not respond to the therapies and developed an aggressive tumor within a short time interval. Patients’ data, including age, sex, isocitrate dehydrogenase (*IDH*) status, tumor volume, tumor location, radiation therapy, use of chemotherapy, and progression-free survival time (PFST) were retrospectively reviewed. All patients involved in the study signed informed consent forms. The methods in the study were in accordance with the guidelines of the World Medical Association Declaration of Helsinki. This study was approved by the Ethical Committee of Bezmialem Vakif University (approval no: 2019-12/14).

### 2.2. DNA Isolation

Genomic DNA was isolated from frozen tumor specimens using PureLink Genomic DNA mini kit (Invitrogen, Waltham, Massachusetts, USA), as described in the manufacturer’s protocol. Approximately 25 mg of each tumor tissue was minced and incubated with 180 µL digestion buffer and 20 µL of Proteinase K at 55 °C for 2 h. The cell lysates were added with 650 µL binding buffer and 100% ethanol. The cell lysates were then loaded to PureLink spin columns and centrifuged at 10,000× *g* for 1 min. The columns were washed with 500 µL of washing buffer and centrifuged at 10,000× *g* for 1 min. Finally, the DNA samples were eluted in 50 µL elution buffers in 1.5 mL Eppendorf. All of the DNA samples were analyzed for quantity and purity by Nanodrop 2000c and sent for sequencing.

### 2.3. Whole Exome Sequencing

Double-stranded DNA capture baits against approximately 36.5 Mb of the human coding exome (targeting > 98% of the coding RefSeq and Gencode v28 regions) were used to enrich target regions from fragmented genomic DNA, using the Twist Human Core Exome Plus kit (Twist Biosciences, San Francisco, CA, USA). The generated library was sequenced on an Illumina HiSeq 2500 platform.

### 2.4. Variant Calling and Variant Annotation

The generated FASTQ files were checked for their quality using FastQC and aligned to the Human Reference Genome Consortium build 37 (hg19) using the Burrows–Wheeler alignment (BWA) algorithm (v. 0.7.17) [21,22,23]. The resulting BAM files were sorted using Samtools (v. 1.7) and visualized using the Integrative Genomics Viewer (IGV) [24]. Variant calling was carried out using Bcftools (v. 1.7) [25]. Variant annotation was performed using the web version of ANNOVAR [26]. Variants were prioritized based on allele frequency and variant prediction scores obtained from ANNOVAR. Allele frequency scores were obtained via ANNOVAR from the 1000 Genomes Project (1000G), Exome Aggregation Consortium (ExAC), Genome Aggregation Database (gnomAD), and Exome Sequencing Project (ESP), and the scores were used for prioritization [27,28,29,30]. Variant prediction scores were obtained also via ANNOVAR, which used the following tools to predict deleterious single nucleotide variants (SNV): Sorting Intolerant from Tolerant (SIFT), Polymorphism Phenotyping (PolyPhen), MutationAssessor, MutationTaster, Combined Annotation Dependent Depletion (CADD), Functional Analysis through Hidden Markov Model (FATHMM), and Likelihood Ratio Test (LRT) [31,32,33,34,35,36,37,38]. Additionally, annotated variant files were converted to Mutation Annotation Format (MAF) using the Maftools package from R programming language to visualize the landscape of critical mutations [39]. Mutated genes identified from both the Maftools analysis and ANNOVAR variant prioritization were queried against four glioblastoma cohort study NGS datasets from the cBioPortal for Cancer Genomics Database [40]. Separately, signaling pathways associated with the mutated genes were analyzed by use of the Protein Analysis Through Evolutionary Relationships (PANTHER) classification system [41].

## 3. Results

### 3.1. Patients

Five glioblastoma tumor samples from four patients who showed rapid re-growth and/or resistance to therapies were studied. The median age of patients was 42 years (range: 19–43 years). The median tumor local progression time was 3 months (range: 1–4 months).

#### 3.1.1. Patient 1 (G1)

A 19-year-old female patient presented with a seizure and was found to have a large frontotemporal heterogeneously contrast-enhancing lesion in the magnetic resonance image (MRI) of the brain (Figure 1a). The patient underwent gross total resection of the lesion (Figure 1b) and histopathological evaluation revealed IDH wild-type glioblastoma WHO grade 4. MRI of the brain showed recurrence three (3) months after the surgery (Figure 1c).

#### 3.1.2. Patient 2 (G2)

A 42-year-old female patient presented with headache and nausea. MRI of the brain showed a partially enhancing lesion in the left frontoparietal lobe (Figure 2a). The patient underwent gross total resection of the tumor (Figure 2b) and histopathological examination showed IDH wild-type glioblastoma WHO grade 4. The patient received radiotherapy with concomitant and adjuvant Temozolomide treatment. After three months, a follow-up MRI of the patient revealed recurrence at the resection cavity (Figure 2c).

#### 3.1.3. Patient 3 (G3–4)

A 42-year-old male patient presented with a headache and an MRI of the brain showed a non-enhancing lesion, consistent with a low-grade glial tumor, in the right frontal region (Figure 3a) and a small enhancing lesion in the left frontal lobe (Figure 3b). The patient opted to wait without any treatment. One month later, an MRI of the brain showed a rapid increase in the size of the right frontal region with possible differentiation to a higher grade (Figure 3c). The patient underwent gross total resection of the right frontal lesion (G3 lesion) and histopathological evaluation revealed IDH wild-type glioblastoma WHO grade 4 (Figure 3d). The patient then received radiotherapy for both the right frontal lobe surgical cavity and the lesion in the left frontal lobe. Thirteen months after the initial surgery, a follow-up MRI of the brain revealed the progression of the tumor in the left frontal lobe and this lesion was subsequently removed (G4 lesion) (Figure 3e,f, respectively). Histopathological evaluation showed IDH wild-type glioblastoma WHO grade 4. Two months after the second surgery for the left frontal lesion, a recurrence was observed at the surgical cavity in the follow-up MRI (Figure 3g).

#### 3.1.4. Patient 4 (G5)

A 43-year-old female patient presented with a speech problem and was diagnosed with a left temporal lobe lesion (Figure 4a). The patient underwent resection of the lesion, which was consistent with IDH wild-type glioblastoma WHO grade 4 (Figure 4b). A follow-up MRI revealed a recurrence of the lesion 2.5 months after the surgery (Figure 4c).

### 3.2. Whole Exome Sequencing and Variant Prioritization

Whole exome sequencing (WES) was performed on the DNA from five (5) tumor tissue samples. The sequencing was observed to be of at least 20× coverage depth for >98% of the targeted bases. A standard bioinformatics pipeline for WES analysis was implemented to identify the somatic mutations in the samples by comparing them with a reference genome (Figure 5).

### 3.3. Mutational Landscape of Rapid Progressive Glioblastoma

Collectively, across all samples, variant annotation through ANNOVAR identified 103,935 genetic alterations (median: 23,081), which included synonymous, missense, insertions, deletions, frame-shift insertions, frame-shift deletions, stop-gain, and nonstop genetic alterations (Figure 6a). Overall, a total of 46,751 missense mutations accounted for ~44.97% of total genetic alterations (median: 10,396).

The variants from ANNOVAR were then evaluated for allele frequencies obtained from 1000G, ExAC, gnomAD, and ESP databases. Given that aggressive glioblastoma is a rare cancer disease [42], identification of variants focused on those that were rare or not observed in the databases. A variant frequency of less than 0.01 was designated as a rare variant by each of the databases. Only variants that met this threshold across each of these databases were selected for further phenotypic evaluation. We used the overall allele frequency value (indicated with the “ALL” tag or likewise) provided by each of the respective databases studied herein. Using the overall value is a limitation herein when compared to using the frequencies specific to a Turkish population, which was not adequately represented in the databases. The phenotypic nature of the resulting selected variants was predicted by use of SIFT, PolyPhen, MutationAssessor, MutationTaster, CADD, FATHMM, and LRT scores (all via ANNOVAR). Given the limited sample size, a consensus variant annotation approach was used to increase the reliability, whereby missense mutations were annotated to be putative deleterious only when identified as such by all of the tools used. A total of 81 somatic alterations from the five samples (with various degrees of overlap between the samples), predicted to be deleterious, met the criterion (Appendix A). These 81 alterations were mapped to a list of 60 unique genes. Notably, among the 81 alterations, only the mitochondrial carrier homolog 2 (*MTCH2*) gene mutation (Chr11: 47647265 A>G) was observed to be common (overlapped) across all the five tumor samples studied (Figure 6b).

The initial somatic mutations (103,935) annotated by ANNOVAR were also analyzed using Maftools, which provides feature-rich customizable visualizations. Overall, the tool identified 49,780 somatic mutations in the five samples (not considering synonymous mutations), which included frame-shift insertions, frame-shift deletions, in-frame deletion, in-frame insertion, missense, nonsense, nonstop, and translation start site mutations (Appendix A). A landscape of the WES analysis generated by plotting the summary of the resulting output MAF files is illustrated in Figure 7. The majority of the variations were missense mutations, and these were categorized as single nucleotide polymorphisms (SNPs), as shown in Figure 7a,b.

The tumor mutation burden (TMB; herein, in the context of number of variants (excluding synonymous mutations)) within each sample is depicted in Figure 7c. The highest numbers of mutations were observed in G1 and G2 samples. Sample G5 showed the least number of mutations. Samples G3 and G4 were tumor tissues from the same patient, where sample G3 demonstrated a higher TMB compared to G4. Overall, 8,468 genes were found to be mutated and among them, the top ten genes with the highest number of mutations were *MUC3A*, *MUC4*, *MUC6*, *OR4C5*, *PDE4DIP*, *AHNAK2*, *OR4C3*, *ZNF806*, *TTN*, and *RP1L1* (Appendix A). The mutation landscape in G3 and G4 was also compared and, although a notable difference was observed in the overall TMB, there was no significant difference (Appendix A). The transition-transversion (TiTv) plots were generated using Maftools to better understand the associated tumor mutation signature. All tumor tissues presented high-intensity transition mutations (Figure 7a,e,f). Among the transition mutations, cytosine to thymine (C>T) was the highest.

### 3.4. Gene Mutation in Glioblastoma of Rapid Progression, Cross-Compared to TCGA Datasets

The gene mutations identified herein were cross-examined with gene mutation datasets of four different cohort studies for glioblastoma from TCGA using cBioPortal (Appendix A). Among the ten most mutated genes identified by Maftools, *MUC3A* showed the highest mutation frequency among our samples; however, it exhibited a low frequency (~5%) in glioblastoma TCGA data (Figure 8). Although *TTN* gene mutations were less frequent in our samples, their mutation frequency was ~18% in glioblastoma TCGA data. This may be explained by the fact that the patients in our cohort had a short recurrence time and the tumor was of aggressive behavior. Separately, the 81 somatic alterations identified via ANNOVAR, which were predicted to be deleterious and rare in the general population, and mapped to a unique list of 60 genes, were also cross-examined with glioblastoma datasets from TCGA using cBioportal (Figure 9). The *MTCH2* gene mutation was predicted to be highly deleterious; however, its mutation frequency in glioblastoma TCGA data was significantly low (~0.6%).

### 3.5. Signaling Pathways Associated with Gene Mutations in Glioblastoma of Rapid Progression

We applied the PANTHER classification system, which incorporates PANTHER/X ontology for gene function classification and ranks non-synonymous single nucleotide variants according to their likelihood of affecting protein functions. This was to characterize the association between the genes (exhibiting prioritized mutations) and signaling pathways, to gain insight into possible underlying signaling mechanisms. All genes containing the 81 selected somatic mutations (deleterious and rare in the general population) identified through ANNOVAR were evaluated for association with signaling pathways. A total of 12 genes were annotated to be involved in 10 signaling pathways (Figure 10a). Similarly, somatic mutations identified using Maftools were also evaluated (only genes containing the top 99 somatic mutations). A total of 18 genes were annotated to be involved in nine (9) cellular signaling pathways (Figure 10b).

## 4. Discussion

Glioblastoma, a rare cancer, is the most malignant central nervous system tumor and is associated with a poor patient prognosis. Despite recent advancements, in chemotherapy and radiotherapy, which increased the OS to 20 months, the patient prognosis remains grim [43]. Intrinsic intra-tumoral heterogeneity leading to aberrations in multiple cellular pathways is attributed to the aggressive nature of glioblastoma [44]. In this study, the characteristics of glioblastomas of rapid progression were examined by use of WES to gain insight into the molecular and mutational signature of aggressive glioblastoma tumors.

Herein, *MTCH2* (Chr11: 47647265 A>G) gene mutation was identified to be potentially associated with highly progressive glioblastoma. Moreover, a high transition (relative to transversion) mutation signature was observed, besides the highest frequency of mutations exhibited by the genes *MUC3A*, *MUC4*, *MUC6*, *OR4C5*, *PDE4DIP*, *AHNAK2*, *OR4C3*, *ZNF806*, *TTN*, and *RP1L1*. However, according to the cBioPortal TCGA data, these gene mutations were present in a low number of publicly available samples. This discrepancy could be attributed to either the heterogeneity of the study population and/or our small sample size.

A total of 81 somatic gene alterations (with varying degree of overlap) across five tumor samples herein from four patients with rapid progression were identified to be putative deleterious and rare, relative to common human genetic variations. The *MTCH2* (Chr11: 47647265 A>G) gene mutation, predicted to be deleterious, was the only one that was common across the five samples, and may likely be associated with the rapid progression of glioblastomas. *MTCH2* is expressed on the outer mitochondrial membrane and regulates cell apoptosis by interacting with truncated BH3-interacting domain death agonist (tBID) [45]. *MTCH2* expression has been associated with several cancers [46,47]. Recently, Yuan et al. explained the oncogenic role of altered *MTCH2* expression in malignant glioma [48]. *MTCH2*, present on the outer mitochondrial membrane, plays a crucial role in chemoresistance and tumor migration in gliomas. In human glioma cells, *MTCH2* knockdown increases the mitochondrial OXPHOs, which subsequently enhances oxidative damage in glioma cells. Abrogation of *MTCH2* gene expression attenuated the invasive property of glioma cells and enhanced Temozolomide sensitivity via down-regulation of the AKT signaling pathway [48].

The *MTCH2* gene mutation (Chr11: 47647265 A>G) is reported in the Single Nucleotide Polymorphism database (dbSNP; rsID-rs76666113; accessed as of December 2021) and has been annotated to result in amino acid change at position 237, from non-polar phenylalanine to polar serine. This point mutation may lead to a net alteration in physiochemical and structural attributes of the *MTCH2* protein, which may lead to an altered function(s) and interaction(s) of the protein. However, a literature search showed no available report, categorically linking the specific gene mutation to glioblastoma. Further, a review of the TCGA datasets showed that the given *MTCH2* gene mutation is of low frequency. Thus, it is unclear whether to implicate or neglect the role of the given *MTCH2* gene mutation in the progression of aggressive glioblastoma, in particular when the datasets assessed through cBioPortal included only those from primary tumor patients. The longitudinal heterogeneity in glioblastomas cannot be neglected, and the lack of TCGA datasets for highly aggressive glioblastoma make it difficult to make a reliable comparison. Although the mutation was selected because of its rare occurrence (frequency of less than 0.01) in the databases of common human genetic variations (1000G, ExAC, gnomAD, and ESP), its frequency in the TCGA glioblastoma patient sample datasets was expected to be higher. The observed low frequency of the mutation in the TCGA glioblastoma datasets suggests a niche role, and thus a future need to assess the frequency in datasets of aggressive glioblastoma of sufficient sample size. The study herein is of a small sample size, hence the lack of statistical significance to implicate the role of *MTCH2* gene mutation in aggressive glioblastoma.

High TMB (general definition: the number of mutations per megabase) has become popular as a candidate biomarker for selecting cancer patients for immune checkpoint blockade therapy [49]. In glioma patients, high TMB is associated with a poor prognosis [50]. It is assumed that the high number of mutated proteins will generate an increased number of antigenic peptides, leading to enhanced immunogenicity [51]. Recently, low TMB was shown to be associated with more prolonged survival of patients with recurrent glioblastoma, treated with immune checkpoint blockade and recombinant polio virotherapy [52]. The study herein suggests a high TMB, with a possible genetic signature represented by the top 10 mutated genes across the four glioblastoma patients with rapid progression: *MUC3A*, *MUC4*, *MUC6*, *OR4C5*, *PDE4DIP*, *AHNAK2*, *OR4C3*, *ZNF806*, *TTN*, and *RP1L1*. Mutations of a transition nature, such as a high C>T mutation frequency, were more common than transversion. In agreement with this observation, in a cohort of 27 paired samples of glioblastoma, C>T transition mutation signature was reported to be dominant [53].

PANTHER pathway enrichment analysis showed that several signaling pathways were linked to the mutated genes identified in this study by use of ANNOVAR and Maftools. The majority of the signaling pathways are related to cellular metabolisms, and others, including the WNT and FAS signaling pathways, have been linked to glioma pathogenesis [54,55].

This, herein, is a pioneering study constituting a WES approach to reveal genes with differential accumulation of somatic mutations in highly aggressive glioblastoma. However, the study has certain limitations. Firstly, it did not include adjacent normal tissues as controls; the inclusion of which could have provided a better baseline genetic makeup to compare to tumor samples. Secondly, the sample size is small to conclude the clinical significance of the detected mutations. Therefore, further investigation is required with larger cohorts to confirm the findings. This may be done by designing a dynamic study, which includes analyzing the evolution of a recurrent tumor with respect to the primary treatment given to the patient. This may provide insight into designing a better first-line therapeutic strategy. Additionally, the mechanism involved in the possible association of *MTCH2* gene mutation to aggressive glioblastoma may be investigated using mutant model systems for a better understanding.

## 5. Conclusions

This study provided an insight into the molecular makeup of glioblastomas of rapid progression. We identified *MTCH2* (Chr11: 47647265 A>G) gene mutation as putative deleterious in the aggressive tumor studied. Moreover, a possible genetic signature is proposed, comprising of highly mutatedgenes, namely *MUC3A*, *MUC4*, *MUC6*, *OR4C5*, *PDE4DIP*, *AHNAK2*, *OR4C3*, *ZNF806*, *TTN*, and *RP1L1* genes. These findings provide critical initial data for further rational development of genetic screening and diagnostic approaches against aggressive glioblastoma.

## Figures and Tables

**Figure 1 diagnostics-13-01017-f001:**
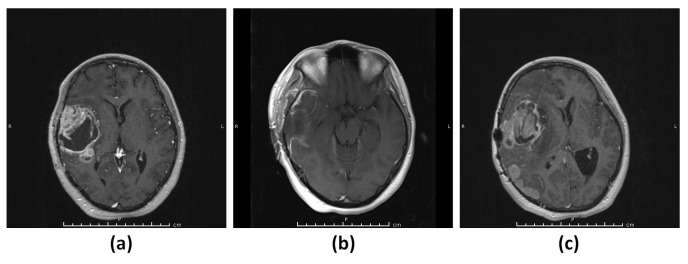
Patient 1 (G1). (**a**) T_1_-weighted MRI showing a contrast-enhancing lesion in the right frontotemporal region. (**b**) T_1_-weighted MRI showing gross total resection of the tumor. (**c**) T_1_-weighted MRI showing rapid recurrence of the tumor.

**Figure 2 diagnostics-13-01017-f002:**
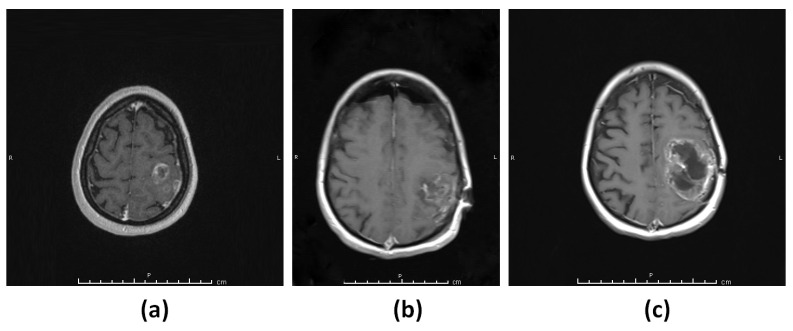
Patient 2 (G2). (**a**) T_1_-weighted MRI showing a contrast-enhancing lesion in the left frontoparietal lobe. (**b**) T_1_-weighted MRI showing gross total resection of the tumor. (**c**) T_1_-weighted MRI showing rapid recurrence of the tumor.

**Figure 3 diagnostics-13-01017-f003:**
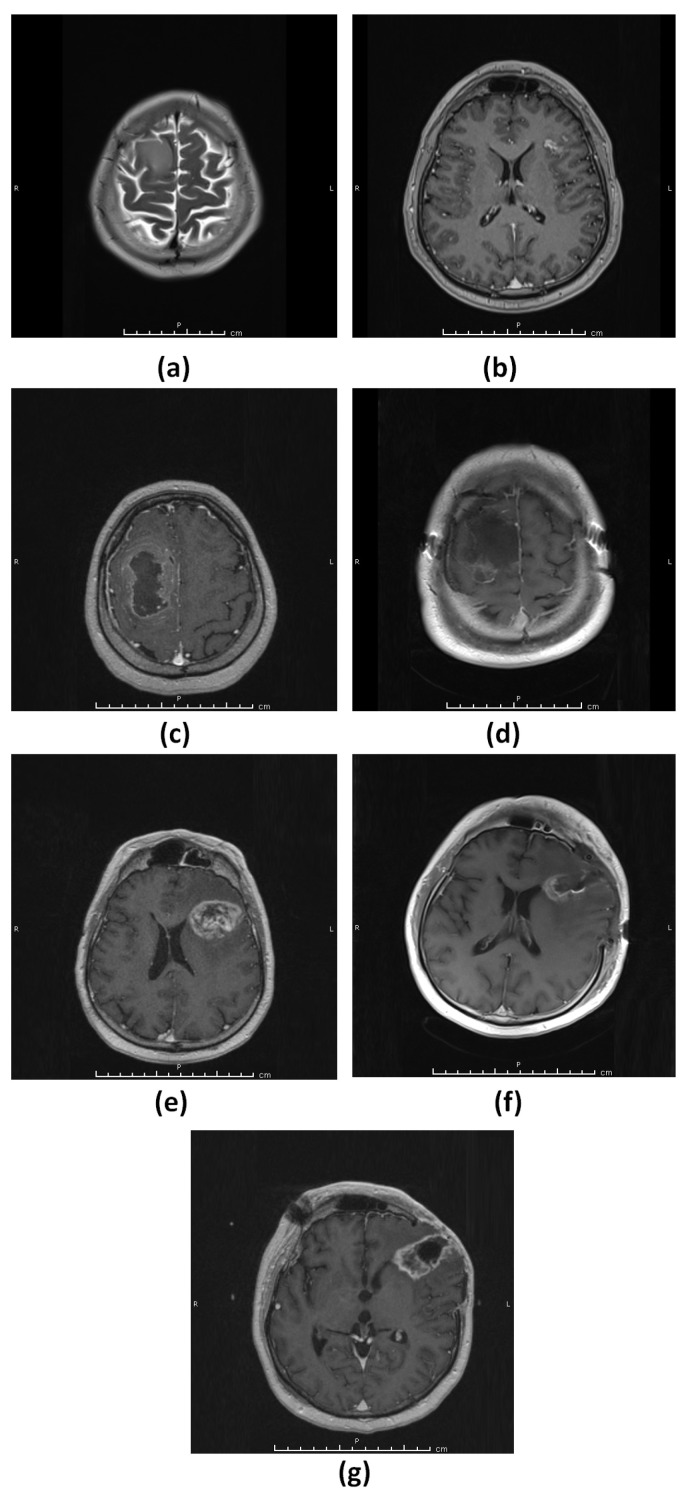
Patient 3 (G3–4). (**a**) T2-weighted MRI showing a hyperintense lesion consistent with a low-grade glial tumor in the right frontal region. (**b**) T1-weighted MRI showing a small contrast-enhancing lesion in the left frontal lobe. (**c**) T1-weighted MRI showing rapid growth and malignant transformation of the untreated right frontal tumor. (**d**) T1-weighted MRI showing gross total resection of the tumor. (**e**) T1-weighted MRI showing the progression of the left frontal tumor. (**f**) T1-weighted MRI showing gross total resection of the tumor. (**g**) T1-weighted MRI showing rapid recurrence of the left frontal tumor.

**Figure 4 diagnostics-13-01017-f004:**
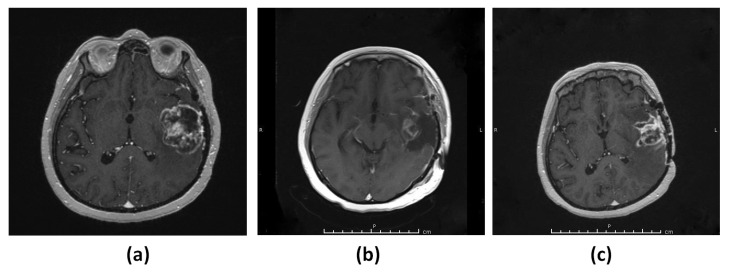
Patient 4 (G5). (**a**) T_1_-weighted MRI showing a contrast-enhancing lesion in the left temporal lobe. (**b**) MRI showing gross total resection of the tumor. (**c**) MRI showing rapid recurrence of the tumor.

**Figure 5 diagnostics-13-01017-f005:**
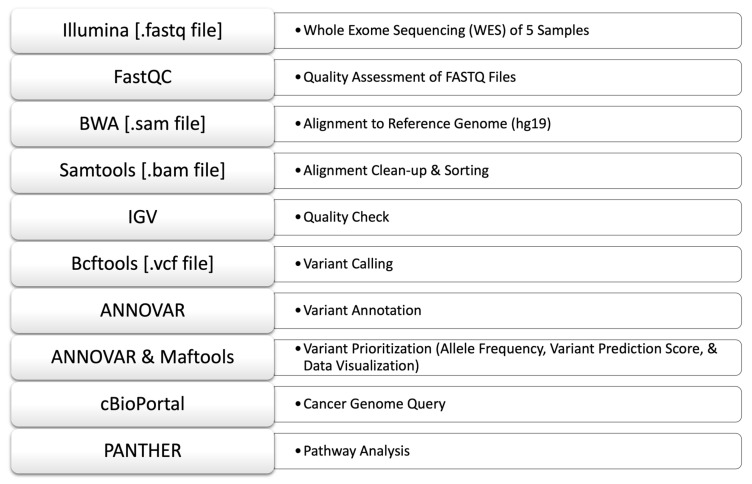
WES data analysis pipeline implemented in this study.

**Figure 6 diagnostics-13-01017-f006:**
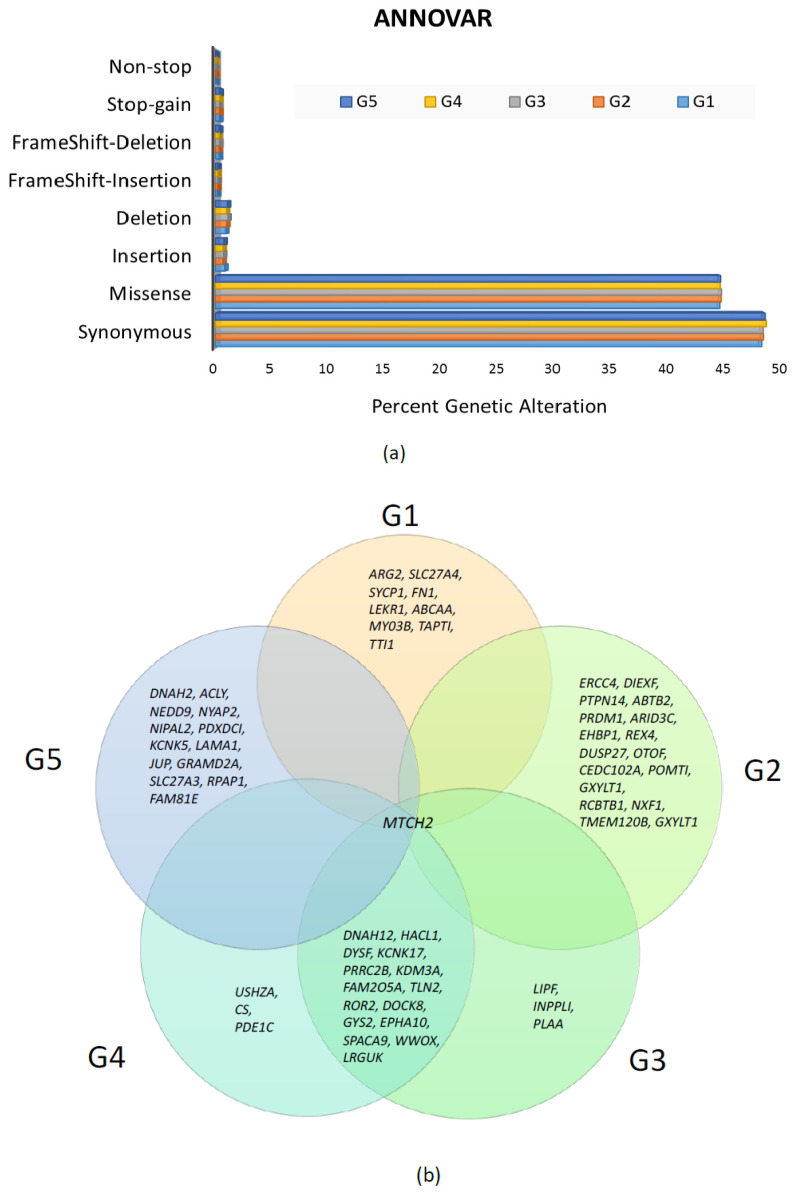
Landscape of genetic alterations by ANNOVAR. (**a**) Percent genetic alteration. (**b**) A Venn diagram showing the genes (of each patient sample) with mutations that met the criteria of being both rare and predicted as deleterious. Samples are indicated with the notations “G1-G2-G3-G4-G5” accordingly. Fifteen genes exhibited mutations that were common between samples G3 and G4 and only *MTCH2* gene mutation was common across the five samples.

**Figure 7 diagnostics-13-01017-f007:**
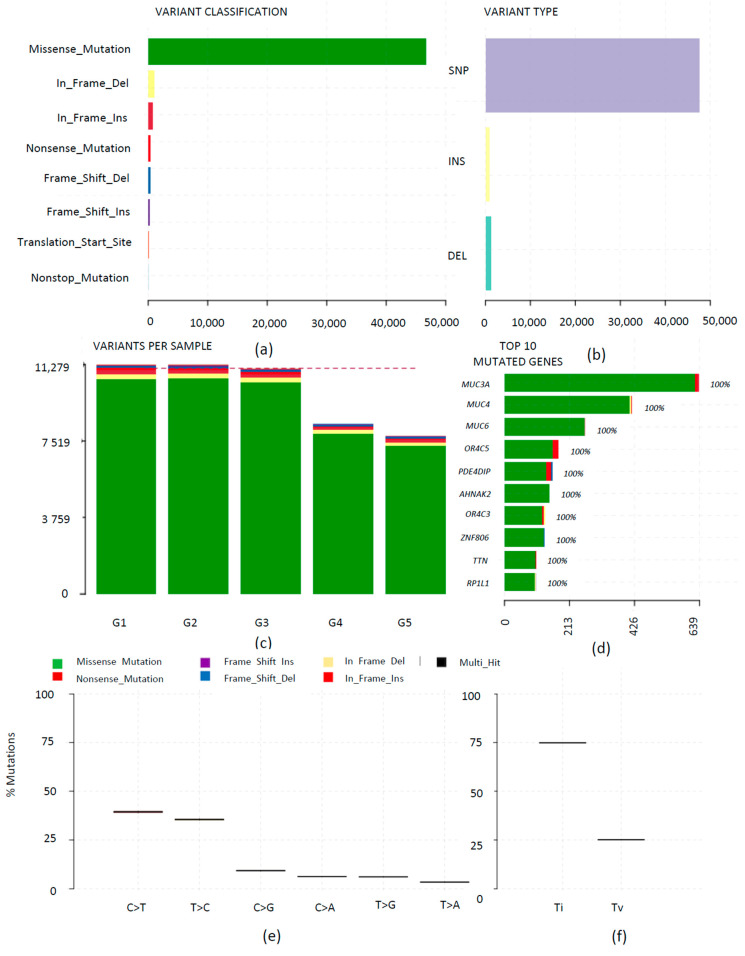
Landscape of 49,780 somatic mutations (not considering synonymous mutations) identified across the five samples by use of Maftools. (**a**) Distribution of different types of mutations. (**b**) The number of different missense mutations. (**c**) Variants observed for each sample. (**d**) Highly mutated genes based on the number of mutations per gene. (**e**) Overall distribution of six different transition/transversion conversions. (**f**) Fraction of transition-transversion (TiTv) conversions.

**Figure 8 diagnostics-13-01017-f008:**
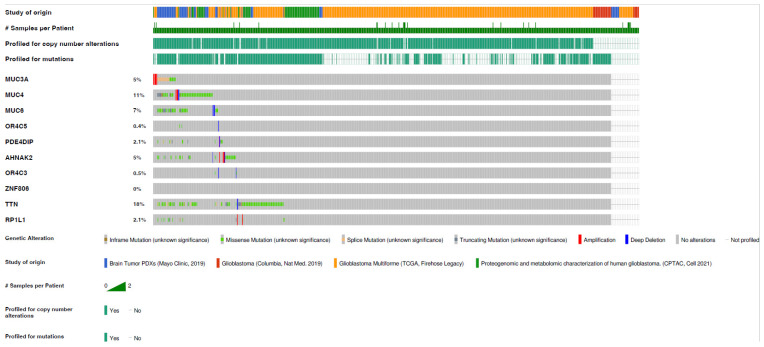
The percentage frequency of mutations in glioblastoma TCGA data for the top ten most mutated genes identified via Maftools from the five aggressive glioblastoma samples studied herein.

**Figure 9 diagnostics-13-01017-f009:**
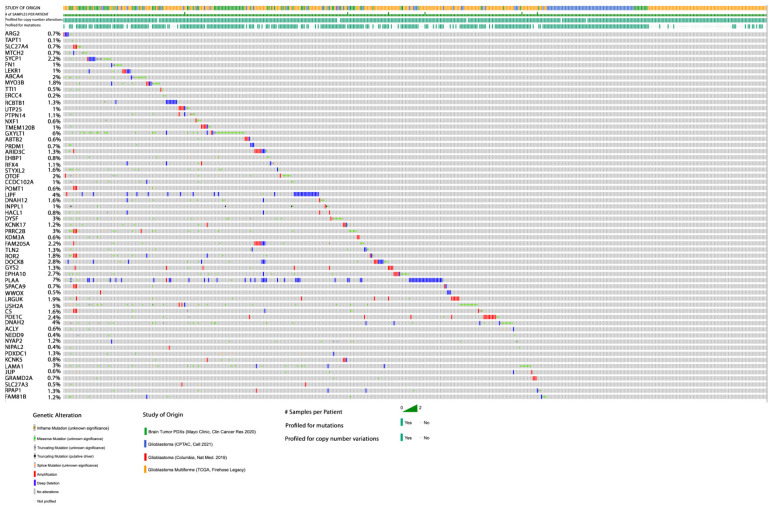
The percentage frequency of mutations in glioblastoma TCGA data for the 60 genes identified via ANNOVAR from the five aggressive glioblastoma samples studied herein. These genes harbored somatic alterations that were both putative deleterious and rare in the general population.

**Figure 10 diagnostics-13-01017-f010:**
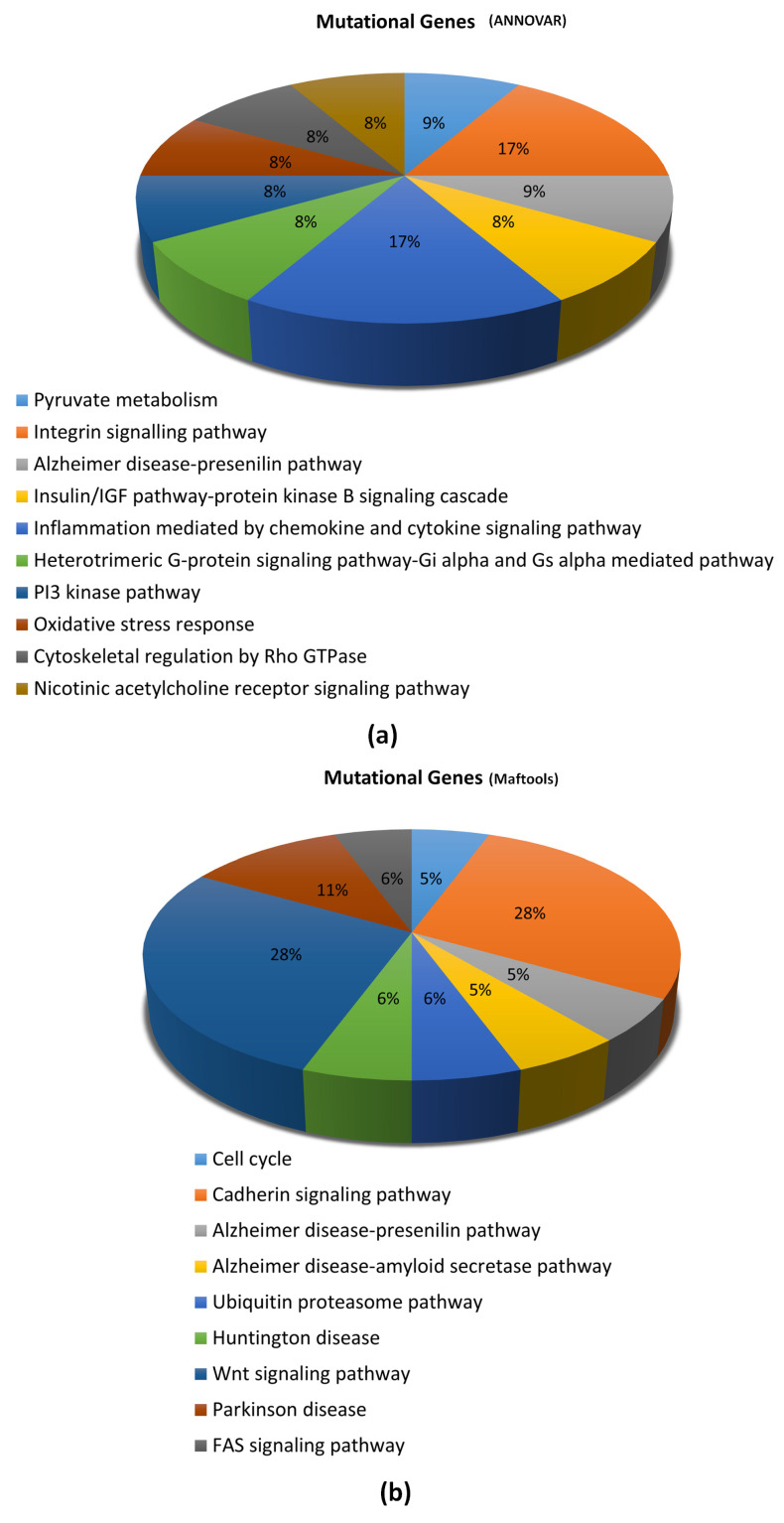
PANTHER analysis of signaling pathways associated with the top mutated genes identified using (**a**) ANNOVAR and (**b**) Maftools.

## Data Availability

The data presented in this study is publicly available at SRA with Accession ID: PRJNA940283 (https://www.ncbi.nlm.nih.gov/bioproject/PRJNA940283, accessed on 5 January 2023).

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
