# Peer review of "Identification of Genetic Alterations in Rapid Progressive Glioblastoma by Use of Whole Exome Sequencing"

_diagnostics, 2023, doi:10.3390/diagnostics13061017_

Round 1
Reviewer 1 Report
The present article "Identification of Genetic Alterations in Rapid Progressive Glioblastoma by Use of Whole Exome Sequencing" written by Khan et al, is a concise and well written article.
Authors need to make following changes to further improve the quality of manuscript.
1)Rewrite the introduction by adding more information to make it understandable for all the readers.
2)The limitations explained in this study regarding present study are valid. However, a section explaining future perspectives can be included with respect to scope of this study.
Author Response
Reviewer 1
Comments and Suggestions for Authors
The present article "Identification of Genetic Alterations in Rapid Progressive Glioblastoma by Use of Whole Exome Sequencing" written by Khan et al, is a concise and well written article.
Authors need to make following changes to further improve the quality of manuscript.
1) Rewrite the introduction by adding more information to make it understandable for all the readers.
Authors Response: We acknowledge the reviewer’s comment and revised not just the introduction accordingly, but the manuscript in general to improve its readability, clarity, and rigour.
2) The limitations explained in this study regarding present study are valid. However, a section explaining future perspectives can be included with respect to scope of this study.
Authors Response: Thanks for the comment; we have added a section about future perspective in the discussion section.
Reviewer 2 Report
The authors perform Whole Exome Sequencing for the Identification of Genetic Alterations in Rapid Progressive Glioblastoma.
The results indicate that MUC3A, MUC4. MUC6, OR4C5, PDE4DIP, AHNAK2, OR4C3, ZNF806, TTN, and RP1L1 genes shows the highest number of substitutions. These findings provide diagnostic approaches against aggressive glioblastoma.
In my point of view, the work is well planned and resolved.
The results and the discussion are clear enough.
Author Response
Comments and Suggestions for Authors
The authors perform Whole Exome Sequencing for the Identification of Genetic Alterations in Rapid Progressive Glioblastoma.
The results indicate that MUC3A, MUC4. MUC6, OR4C5, PDE4DIP, AHNAK2, OR4C3, ZNF806, TTN, and RP1L1 genes shows the highest number of substitutions. These findings provide diagnostic approaches against aggressive glioblastoma.
In my point of view, the work is well planned and resolved.
The results and the discussion are clear enough.
Authors Response: Thank you for the feedback. We took the opportunity to further improve the manuscript in general, to bring about better readability, clarity, and rigour.
Reviewer 3 Report
In this manuscript by Khan et al., the authors aimed to investigate the genetic alteration using the whole exome sequencing of the five aggressive glioblastoma samples. The authors found many genes with missense substitutions, provided the list of 10 genes with the highest number of substitutions, and suggest that MTCH2 gene sequence change may be deleterious in aggressive tumors.
The authors discussed different positive and negative aspects of their findings carefully.
I have the following comments for the authors' consideration-
Major Comments:
- As mentioned in the discussion, the authors have not used any normal tissue or blood sample to know the difference in the genetic background of an individual from the available reference human genome sequence. The number of samples is very low and missense mutations are observed in different sets of genes except for MTCH2 in different samples (G1 to G5). The authors should discuss how a mutation in MTCH2 can contribute specifically to glioblastoma.
- The tumor mutation burden and mutations in G3 and G4 samples which are from the same patient also differ suggesting the heterogeneity and complexity of glioblastoma tumors. This also suggests how difficult it is to conclude these findings for glioblastoma.
Minor Comments
- Please increase the font size in figure 9.
Author Response
Comments and Suggestions for Authors
In this manuscript by Khan et al., the authors aimed to investigate the genetic alteration using the whole exome sequencing of the five aggressive glioblastoma samples. The authors found many genes with missense substitutions, provided the list of 10 genes with the highest number of substitutions, and suggest that MTCH2 gene sequence change may be deleterious in aggressive tumors.
The authors discussed different positive and negative aspects of their findings carefully.
I have the following comments for the authors' consideration-
Major Comments:
As mentioned in the discussion, the authors have not used any normal tissue or blood sample to know the difference in the genetic background of an individual from the available reference human genome sequence. The number of samples is very low and missense mutations are observed in different sets of genes except for MTCH2 in different samples (G1 to G5). The authors should discuss how a mutation in MTCH2 can contribute specifically to glioblastoma.
Authors Response: We acknowledged the reviewer’s comment and have revised the discussion accordingly. Further, we improved the manuscript in general, to bring about better readability, clarity, and rigour.
The tumor mutation burden and mutations in G3 and G4 samples which are from the same patient also differ suggesting the heterogeneity and complexity of glioblastoma tumors. This also suggests how difficult it is to conclude these findings for glioblastoma.
Authors Response: We acknowledged the reviewer’s comment and have further elaborated on the heterogeneity and complexity of glioblastoma tumors. Moreover, we improved the manuscript in general, to bring about better readability, clarity, and rigour.
Minor Comments
Please increase the font size in figure 9.
Authors Response: Font size and resolution have been increased.